# *Echinacea purpurea* and *Onopordum acanthium* Combined Extracts Cause Immunomodulatory Effects in Lipopolysaccharide-Challenged Rats

**DOI:** 10.3390/plants13233397

**Published:** 2024-12-03

**Authors:** Maria Vlasheva, Mariana Katsarova, Ilin Kandilarov, Hristina Zlatanova-Tenisheva, Petya Gardjeva, Petko Denev, Nora Sadakova, Viktor Filipov, Ilia Kostadinov, Stela Dimitrova

**Affiliations:** 1Department of Bioorganic Chemistry, Faculty of Pharmacy, Medical University of Plovdiv, 15A Vassil Aprilov Blvd., 4002 Plovdiv, Bulgaria; mariya.vlasheva@mu-plovdiv.bg (M.V.); stela.dimitrova@mu-plovdiv.bg (S.D.); 2Department of Pharmacology and Clinical Pharmacology, Faculty of Medicine, Medical University of Plovdiv, 15A Vassil Aprilov Blvd., 4002 Plovdiv, Bulgaria; ilin.kandilarov@mu-plovdiv.bg (I.K.); hristina.zlatanova@mu-plovdiv.bg (H.Z.-T.); iliya.kostadinov@mu-plovdiv.bg (I.K.); 3Department of Microbiology and Immunology, Faculty of Medicine, Medical University of Plovdiv, 15A Vasil Aprilov Blvd., 4002 Plovdiv, Bulgaria; petya.gardjeva@mu-plovdiv.bg; 4Laboratory of Biologically Active Substances, Institute of Organic Chemistry with Centre of Phytochemistry, Bulgarian Academy of Sciences, 139 Ruski Blvd., 4000 Plovdiv, Bulgaria; petko.denev@orgchm.bas.bg; 5Clinic of Neurology, St. Panteleimon Hospital Plovdiv, 9 Nicola Vaptsarov Blvd., 4004 Plovdiv, Bulgaria; norasadakova@abv.bg; 6Faculty of Medicine, Medical University of Plovdiv, 15A Vasil Aprilov Blvd., 4002 Plovdiv, Bulgaria; vviktorfilipovv@gmail.com; 7Research Institute, Medical University of Plovdiv, 15A Vassil Aprilov Blvd., 4002 Plovdiv, Bulgaria

**Keywords:** polyphenols, combined extracts, TNF-α, IFN-γ, IL-10

## Abstract

*Echinacea purpurea* and *Onopordum acanthium,* which belong to the Asteraceae family, are widely used plants in traditional medicine. Their antioxidant, anti-inflammatory, antiviral, antibacterial, and antitumor effects are well known. However, there are no data on the effects of their combination. The aim of the present study was to combine *E. purpurea* with *O. acanthium* to study the in vivo immunomodulatory effect of two combinations and to compare it with that of single plants. Their total polyphenolic and flavonoid content and the amounts of individual compounds characteristic of both species were determined. The influence of the obtained extracts on the serum concentrations of cytokines IFN-γ, TNF-α, and IL-10 in experimental animals with lipopolysaccharide-induced systemic inflammatory response was investigated. This research found that a combination of *E. purpurea*/*O. acanthium* in the ratio 1:1 reduced the proinflammatory cytokines TNF-α (244.82 pg/mL) and IFN-γ (1327.92 pg/mL) compared to the LPS-control, respectively, (574.17 pg/mL) and (3354.00 pg/mL), and the combination *E. purpurea*/*O. acanthium* in the ratio of 3:1 significantly increased the levels of the anti-inflammatory cytokine IL-10 (1313.95 pg/mL) compared to the LPS-control (760.09 pg/mL). In conclusion, our results could be a basis for future biomedical research on creating phytopreparations with an immunomodulatory effect.

## 1. Introduction

Immunity is supported by a complex defense system of cells and signaling molecules that work together to protect the body from microbes, viruses, and allergens [1]. The immune system provides immunity through two immune subtypes—innate and adaptive immunity. Innate immunity is a nonspecific immune response that acts as a first line of defense and rapidly responds to the entry of foreign particles through monocytes, neutrophils, dendritic cells, and macrophages. Adaptive immunity is a specific immune response and includes B and T lymphocytes that release antigen-specific antibodies and cytokines. They work together, and when defects or overstimulation occur, the immune system becomes dysfunctional. In such cases, individuals with an inadequate immune response are at an increased risk of developing life-threatening infections and chronic diseases such as allergies, autoimmune diseases, and cancer. In this regard, immunomodulatory therapy has attracted attention in recent years [2]. It is based on endogenous and exogenous factors that could influence the efficiency of the immune system through immunostimulation or immunosuppression [3].

It is well known that a number of medicinal plants that are used in traditional medicine due to their pharmacological action also have immunomodulatory properties [4]. These plants have been shown to maintain immune homeostasis by regulating signaling pathways involved in oxidative stress responses and inflammatory processes, often leading to chronic inflammation, autoimmune diseases, and cancer. The immunomodulatory properties of medicinal plants are determined by the ability of their biologically active substances to communicate B and T lymphocytes, macrophages, dendritic cells, cytokines, transcription factors, and their downstream signaling pathways, which are responsible for the immune response [5]. Medicinal plants modulate the functions of the immune system, both through a direct effect on immunocompetent cells and indirectly through influencing the production of pro- and anti-inflammatory molecules [6]. Naive CD4 T cells differentiate into several cell types (Th1, Th2, and Th17) based on the cytokines they produce [7]. IFN-γ, TNF-α, IL-1β, IL-2 and IL-12 are Th1 cytokines [8]. They have a proinflammatory effect, which, in the absence of adequate control, can lead to tissue damage, such as in autoimmune diseases. IL-4, IL-5, IL-13 and the anti-inflammatory IL-10 are Th2 cytokines. The normal course of the immune reaction is closely dependent on the balance between the Th1 and Th2 immune response [9,10]. The question arises whether the medicinal plants that influence the functions of the immune system contribute to the restoration and maintenance of this balance or disturb it at the expense of stimulating only the Th1 or Th2 immune response. The latter would be the reason for the development of adverse reactions due to an uncontrolled stimulation of the immune system.

The immunomodulatory properties of active substances and extracts from traditional plants correlate with their antioxidant, antiallergic, antiproliferative, anti-inflammatory, and antitumor activities [11]. Some of the most extensively studied plants from this point of view are *Curcuma Longa* L., *Panax ginseng* CA Meyer, and *Moringa oleifera* Lam [2]. Apart from them, *E. purpurea* and *O. acantium*, which are medicinal plants belonging to the *Asteraceae* family, are also broadly used in traditional medicine. A number of in vitro and in vivo studies show the wide range of effects they possess. As for *E. purpurea*, the anti-inflammatory, antimycotic, antibacterial, antiviral, immunomodulatory [12,13], as well as antitumor, hepatoprotective, or neuroprotective [14] effects of its extracts have been proven. Extracts of *O. acantium* exhibit anti-inflammatory, antitumor, and cardiostimulating effects [15], as well as antibacterial [16], analgesic, and antipyretic [17] activities. Undoubtedly, these effects are determined by the biologically active substances that plants synthesize in the course of their development. The *Echinacea* genus is characterized by polyphenols (caffeic acid derivatives, flavonoids, alkylamides, polysaccharides [14], and the *Onopordum* genus by polyphenols (flavonoids, lignans, phenolic acids) [18], terpenes and steroids [19].

Previous studies have shown that the use of *Echinacea* extract can reduce the severity and duration of respiratory tract infections [13]. However, during the COVID-19 pandemic, there have been concerns that its ability to stimulate the immune system may contribute to or enhance the potential for a cytokine storm [20]. Also, some adverse effects of *Echinacea* extract are known, such as abdominal discomfort, bronchospasm, hypersensitivity reactions, headaches, dizziness, drowsiness, and possible interactions with conventional drugs [21]. It is not recommended for patients who have leukemia, tuberculosis, multiple sclerosis, autoimmune diseases, and HIV infections [22].

The present work aimed to combine *E. purpurea* with *O. acanthium*, to study the in vivo immunomodulatory effect of the combinations, and to compare them with that of the single plants. In this regard, water-ethanol extracts of *E. purpurea, O. acanthium*, and two combinations in the ratio 1:1 and 3:1, respectively, of *Echinacea and Onopordum* were prepared. We suggest combined extracts of *E. purpurea* and *O. acanthium* for the first time. The ratio between the drugs 1:1 was chosen, similar to the combination of *Echinacea* and *Hypericum perforatum* studied by Bajrai et al. [23], while the ratio 3:1 is entirely our proposal, following the idea of keeping the effect of *E. purpurea* but eliminating possible side effects. Their total polyphenolic and flavonoid content and the amounts of individual compounds characteristic of both species were determined. The influence of the obtained extracts on the serum concentrations of the Th1 cytokines IFN-γ and TNF-α and the Th2 cytokine IL-10 in experimental animals with lipopolysaccharide (LPS) induced systemic inflammatory response was investigated.

## 2. Results

### 2.1. Determination of Individual and Combined Plant Extracts Composition

A preliminary analysis was conducted to determine the total polyphenolic and flavonoid contents of the ethanolic extracts of *E. purpurea*, *O. acanthium*, and their combined extracts at the following ratios of 1:1 *E. purpurea* to *O. acanthium* (Combination 1), and 3:1 *E. purpurea* to *O. acanthium* (Combination 2). These results are presented in Table 1.

The chromatographic determination of individual compounds characteristic of both plants was made; phenolic acids-ferulic and caffeic, derivatives of caffeic chicoric acid, caftaric, chlorogenic, neochlorogenic acids, cynarin, echinacoside, flavonoids-quercetin, apigenin, rutin, myricetin, epicatechin, scutellarin, and arctigenin were found (Figure 1). The correlation coefficients calculated between the spectra of standards for all these substances and the spectra of these compounds in tested samples were between 0.9977 and 0.9998 (Table 2). This way, the spectral similarity of the peaks of the substances of interest was estimated. Their UV spectra are shown in Appendix A. The determined amounts of these substances are shown in Table 3.

*Echinacea purpurea’s* extract contained the highest amounts of chicoric, caftaric, and caffeic acids. A greater number of phenolic acids and caffeic acid derivatives were determined in the single extract of *E. purpurea*, and, accordingly, their amount was greater compared to that in the extract of *O. acanthium*. The single extract of *O. acanthium* contained a greater number of the flavonoids studied, but their amount was less than that of the extract of *E. purpurea*. The flavonoid content in this extract was represented by myricetin, quercetin, and apigenin in the highest amounts. Arctigenin is characteristic of *Onopordum* species, and its amount in the studied extract was 555.0 μg/g. Logically, in the combined extracts Combination 1 and Combination 2, almost all the sought-after substances are present, and their amounts are proportionally distributed. The chromatograms of extracts obtained by HPLC are shown in Appendix A.

In our experiment, we used 48 Wistar rats divided into six groups (Table 4). After the treatment with the extracts, a systemic inflammatory response was induced by an intraperitoneal injection of LPS on day 15 of the experiment. Blood samples for immunological assays were collected 4 h after LPS injection. The serum cytokines were measured, and their concentrations in the experimental groups were compared with those of the control groups.

### 2.2. Changes in Serum TNF-a in LPS-Challenged Rats

The serum level of TNF-α in the LPS-treated animals was significantly increased compared to that in the untreated control animals. In all other groups, the levels of the investigated cytokines were decreased compared to the control treated with lipopolysaccharide. The decrease in serum TNF-α concentration in the Combination 1-treated group was most pronounced (244.82 ± 80.99 pg/mL) compared to the LPS-control (574.17 ± 97.96 pg/mL), while in the Combination 2-treated group (322.46 ± 45.61 pg/mL) it was not as large. Both results were statistically significant, *p* < 0.05. The effect of single extracts on the TNF-α levels was not pronounced, although in animals treated with *O. acanthium,* the reduction of the studied cytokine compared to the control treated with lipopolysaccharide was statistically significant. These results are shown in Figure 2.

### 2.3. Changes in Serum IFN-γ in LPS-Challenged Rats

In the control group of animals treated with LPS, the level of the proinflammatory cytokine IFN-γ increased significantly compared to that of the control group not treated with LPS. In the groups treated with the studied combinations and individual extracts, the values of IFN-γ decreased. The decrease in the levels of the investigated cytokine was statistically significant in Combination 1 as well as in the *Onopordum* extract alone, while in Combination 2 and in the *Echinacea* extract alone, a downward trend was observed without statistical significance. The decrease in serum levels of IFN-γ was most pronounced in the animals of the group treated with Combination 1 compared to the control group treated with LPS. If Combination 1 was compared to Combination 2 as well as to *Echinacea* extract alone, the decrease in IFN-γ concentration was also statistically significant. These results are shown in Figure 3.

### 2.4. Changes in Serum IL-10 in LPS-Challenged Rats

The concentration of IL-10 in the serum of rats from the control group treated with LPS increased with statistical significance compared to the untreated control group. When compared with the control group treated with LPS, IL-10 levels increased in all other groups treated with the tested extracts and combinations. In Combination 2, the difference was statistically significant. These results are presented in Figure 4.

### 2.5. Changes in Serum IFN-γ/IL-10 Ratio in LPS-Challenged Rats

In all groups treated with the investigated extracts and in the LPS control group, an increase in the ratio of IFN-γ/IL-10 was observed when compared to the untreated control. This increase was most pronounced in the LPS-treated control group (4.51 ± 0.18). In animals from both groups treated with the combined extracts, a reduction in this ratio was observed when compared to the LPS control group (*p* < 0.001 and *p* < 0.05, respectively, for Combination 1 and Combination 2). A greater reduction was recorded in rats treated with combination 1 (1.55 ± 0.8). A statistically significant decrease in IFN-γ/IL-10 was also observed when comparing Combination 1 with the individual extract (*p* < 0.05). The results are presented in Figure 5.

## 3. Discussion

The use of herbal preparations based on *Echinacea* has an undeniable effect on health in several aspects. However, data on combinations of *Echinacea* with other medicinal plants are scarce. During the COVID-19 pandemic, many attempts have been made to systematize knowledge about the immunostimulatory/immunomodulatory action of *Echinacea* and its benefits in the fight against the virus [20,22] or in immunosuppressed conditions [24]. Bajrai et al. examined the antiviral and virucidal effects of a combination of *Echinacea* and *Hypericum perforatum* against SARS-CoV-2 in vitro [23]. The idea to combine *E. purpurea* and *O. acanthium* arose from the fact that despite the many positive effects of *Echinacea*, some side effects still exist [21]. On the other hand, *O. acanthium* is considered a major agricultural and wild growing weed, unpretentious to the conditions of development, and, accordingly, easily accessible [25,26], but rich in bioactive substances [18,19]. Since water-ethanol extracts have been shown to be the richest in polyphenolic compounds [27,28], such extracts from aerial parts of *E. purpurea* and flowers of *O. acanthium* were used for our study. The amounts of polyphenols that were determined in the *E. purpurea* extract (3843.4 mg GAE/100 g) were comparable to those obtained in the authors’ previous study on *E. purpurea* (3905.4 to 4493.3 mg GAE/100 g) [29] and twice as much compared to *E. purpurea* root extract (1500 mg GAE/100 g) [30]. In the *E. purpurea* extract, the species-specific chicoric and caftaric acids were determined to be 12,915.7 and 3060.0 μg/g, respectively, which are comparable to the amounts obtained by Temerdashev et al. [31]. From the group of flavonoids, rutin was found in the highest amount of 2300.0 μg/g, which is seven times more compared to the result of the same research team [31]. As for the extract of *O. acanthium,* the total polyphenols were 1052.2 mg GAE/100 g, while Parzhanova et al. reported 16,800.0 mg GAE/100 g, but myricetin was in a comparable amount to what they found [28]. These comparisons were made only as an illustration of the fact that it is necessary to standardize any extract with which in vitro or in vivo experiments are conducted on the plant-specific substances responsible for its effect. Of course, their quantities always vary according to the habitat and the conditions of development [29]. Our authors’ team has experience in the creation of combined extracts in which active compounds from different classes are combined, and accordingly, their effects exceed those of individual plants [32].

In the present study, the immunomodulatory effect of *E. purpurea* and *O. acanthium* and their combinations were studied in relation to the LPS-induced systemic inflammatory response. Our results demonstrate the stimulatory effect of bacterial LPS on the production of TNF-α, INF-γ, and IL-10. In the animals from the LPS control, a significant increase in their serum levels was found compared to the pure control. The increase of IL-10 is related to its role as an anti-inflammatory molecule, which is important for preventing the consequences of the intense inflammatory response. IL-10 suppresses the production of TNF-α, IL-1, IL-6, IL-8 formed in response to inflammation. IL-10 synthesis is inhibited by microRNA-98 (miR-98). Under conditions of LPS-induced inflammation, a decreased expression of this microRNA was observed, which may explain the increased production of this anti-inflammatory cytokine [33].

The administration of *E. purpurea*-only extract decreased serum concentrations of TNF-α and INF-γ in LPS-treated animals, but significance was not reached. Data regarding the effect of *E. purpurea* on TNF-α expression are contradictory in the scientific literature. Earlier in vitro studies showed that *E. purpurea* stimulated the production of proinflammatory cytokines, including TNF-α, by human macrophages [34,35]. In more recent in vitro experiments on splenic lymphocytes of rats treated with *E. purpurea*, Yamada et al. found that *E. purpurea* significantly increased the production of IL-2 and INF-γ (especially in the presence of inflammatory stimulators such as LPS and concanavalin A), but not that of TNF-α [36]. In the conditions of the LPS stimulation of macrophages, *E. purpurea* extract leads to a significant decrease in the levels of NF-κB and TNF-α [37]. In an in vivo model of chronic inflammation, the essential oil of *E. purpurea* flowers reduced the level of proinflammatory cytokines (IL-2, IL-6. TNF-α) in the serum [38]. Although TNF-α has a key role in the normal course of the immune response, its overproduction leads to a damaging effect and is important for the pathogenesis of a number of autoimmune diseases, such as rheumatoid arthritis, inflammatory bowel diseases, and psoriasis, among others. [39]. Uncontrolled production of INF-γ can also lead to tissue damage and play a role in disease development [40,41]. INF-γ is able to induce the synthesis of other inflammatory molecules, including TNF-α [42]. The use of immunostimulants, including those of plant origin, might be the cause of the development of autoimmune diseases [43]. Increased and inadequate secretion of TNF-α is one possible explanation for such an adverse reaction [44]. Data regarding the risk of such complications when taking preparations containing *E. purpurea* are scarce [44,45,46]. Most likely, *E. purpurea* and the biologically active substances contained in it have an immunomodulating rather than an immunostimulating effect [22]. The results of the present study support this thesis, as in the animals treated with *E. purpurea* extract, the levels of TNF-α were lower than those of the LPS control. The same was found with respect to the serum concentration of INF-γ, regardless of the data available in the literature for a stimulatory effect of *E. purpurea* on NK cell function [47]. We can assume that in the conditions of a pronounced inflammatory response, the *E. purpurea* extract suppresses the synthesis of proinflammatory molecules in order to limit tissue damage. This is a manifestation of its immunomodulatory effect, which is at least partially realized by increasing the levels of anti-inflammatory mediators, such as IL-10. In in vitro conditions, Gertsch J et al. found that in the presence of LPS, *E. purpurea* suppresses the synthesis of TNF-α. This effect is attributed to the alkylamides contained in the plant, which act as antagonists of the cannabinoid CB2 receptors [48,49]. Phenolic acids (chicoric, caftaric, and caffeic) and flavonoids most likely also contribute to the immunomodulatory effect of *E. purpurea*. Chicoric acid reduced the serum concentration of TNF-α and IL-1β, as well as the mRNA expression of these molecules in the brain of mice with LPS-induced inflammation [50]. In a mouse model of parkinsonism, chicoric acid reduced the level of INF-γ in serum, colon, striatum, and spleen [51]. Of the three phenolic acids, we found that chicoric had the highest concentration in the studied extract. Quercetin, which is also found in the extract subject to the present study, has the ability to inhibit the TLR-4-mediated synthesis of proinflammatory molecules (TNF-α, IL-1β, IL-6, nitric oxide, prostaglandins, etc.) in conditions of LPS-induced inflammation [52]. There is also evidence of a stimulatory effect on INF-γ synthesis, but this was observed in blood mononuclear cells that were not stimulated with a proinflammatory agent [53].

Surprisingly, with *O. acanthium* extract, the lowering effect on serum levels of TNF-α and INF-γ reached statistical significance. Although there is evidence of anti-inflammatory activity in literature [54], this plant is not known as an immunomodulator. Shabsoug B et al. found that *O. acanthium* stimulated NK cell cytotoxicity and increased the production of TNF-α and INF-γ [55]. Recent experimental data indicate a decrease in serum levels of TNF-α by *O. acanthium* in rats with streptozotocin-induced diabetes [56]. The decreased synthesis of TNF-α may be related to the observed inhibition of NF-κB1 expression by *O. acanthium* [57]. The effect of the extract of this plant on the reduced serum level of TNF-α and INF-γ that we recorded can be explained by the biologically active substances contained in it. As mentioned above, quercetin has a pronounced anti-inflammatory effect. A number of studies have demonstrated the ability of arctigenin to suppress the expression of TNF-α and other proinflammatory cytokines—IL-1β, IL-6, etc. [58]. Scutellarin and apigenin also inhibit the production of proinflammatory cytokines and reduce the expression of molecules involved in the inflammatory response [59,60]. The combined administration of the extracts significantly reduced the serum levels of TNF-α and INF-γ, with the effect most pronounced in Combination 1. Combining *E. purpurea* and *O. acanthium* in a ratio of 1:1 leads to the most optimal effect on the level of TNF-α and INF-γ in the conditions of induced systemic inflammation as a result of supplementing and enhancing the action of biologically active substances contained in both extracts. This combination could be useful in the treatment of infectious diseases with a lower risk of overstimulation of the immune system, especially in predisposed patients.

IL-10 suppresses the formation and release of proinflammatory cytokines and directly inhibits certain immune cells. In this way, this cytokine suppresses the development of the inflammatory and immune response. This limits tissue damage [61]. Experimental studies in vitro and in animals demonstrate that *E. purpurea* increases the expression of IL-10 [20]. The results of the present study show that the two examined extracts, as well as their combination in equal amounts, non-significantly increased the serum levels of IL-10. Statistical significance was reached only with Combination 2. This is most likely a compensatory response and is related to the high levels of INF-γ in the animals of this group. IL-10 is known to be the major negative regulator of INF-γ secretion in the setting of LPS-induced inflammation [62].

## 4. Materials and Methods

### 4.1. Chemicals

All analytical standards (ferulic, caffeic, caftaric, chicoric acids, cynarin, echinacoside chlorogenic and neochlorogenic acids, quercetin, apigenin, rutin, myricetin, epicatechin, scutellarin, arctigenin, Folin-Ciocalteu’s reagent, methanol, and acetonitrile (HPLC gradient grade) were purchased from Sigma-Aldrich (Darmstadt, Germany). Water was obtained from a Milli-Q Gradient water purification system (Millipore, Barnstead, NH, USA). Lipopolysaccharide (LPS) was obtained from *E. coli* O55:B5 (Sigma Aldrich, Darmstadt, Germany); Rat TNF-α, INF-γ, and IL-10 ELISA kits (DiaClone, Besançon, France) were used.

### 4.2. Plant Material

Dry aerial parts of *E. purpurea* (*Asteraceae*) (batch number:39397, expiration date: 22 November 2022) and flowers of *O. acanthium* (*Asteraceae*) (batch number: 39398, expiration date: 18 February 2022) were purchased from Herb Pharmacy 36.6 in Plovdiv, Bulgaria and accompanied by a certificate of quality from MediHerb-83 Ltd., Plovdiv, Bulgaria. The dried plant material was powdered, and individual and combined extracts were prepared with the drug ratio *E. purpurea* to *O. acanthium* in a 1:1 (Combination 1) and *E. purpurea* to *O. acanthium* in a 3:1 (Combination 2).

### 4.3. Extraction and Determination of Polyphenols

Approximately 0.5 g of the dried powders were weighed, transferred to extraction tubes, and mixed with 40 mL of the extragent (60% acetone solution in 0.5% formic acid) [63]. The extraction was conducted on an orbital shaker at room temperature for 1 h. Afterward, the samples were centrifuged (6000× *g*) (Hettich EBA 20, Tuttlingen, Germany), and supernatants were used for the analysis of total polyphenols and flavonoids. The total polyphenols were colorimetrically determined with the Folin–Ciocalteu’s reagent according to the method of Singleton et al. [64]. Gallic acid was employed as a calibration standard, and the results were calculated as mg gallic acid equivalents (GAE) per 100 g dry weight. The total flavonoid content was determined with AlCl_3_ reagent, according to Chang et al. [65]. The calibration curve was constructed with quercetin dihydrate (10–200 mg/L). The results were expressed as mg quercetin equivalents (QE) per 100 g dry weight.

### 4.4. Extraction and HPLC Analysis of Individual Compounds

#### 4.4.1. Extraction

The extractions were performed with 60% ethanol since it was found that water-ethanol mixtures optimally extract the phenolic compounds [66,67]. Solutions of powdered raw material with 60% ethanol were prepared in concentrations of two percent. Extractions were performed via maceration at room temperature (25 °C) for 72 h in the dark. The initial extracts were then filtered through a microfilter (0.45 μm) (Merck, Darmstadt, Germany) and injected into the HPLC systems.

#### 4.4.2. HPLC Analysis

The phenolic compounds typical for *E. purpurea* were determined according to the method previously described by Vlasheva et al. [29] using an HPLC system ProStar 230 solvent delivery module and photodiode array detector model 335 (Varian, Belrose, Australia); Hitachi C18 AQ (250 mm × 4.6 mm, 5 μm) column (Hitachi, Tokyo, Japan). This method was modified and used for the separation of chlorogenic acid, caffeic acid, scutellarin, quercetin, arctigenin, and apigenin; solvent system—H_2_O (A) with pH 3.0 and acetonitrile/methanol in a ratio 40:60 (B) in gradient condition from 0 to 9 min. 80A:20B–50A:50B; 9–26 min. 50A:50B–10A:90B; 26–28 min. 10A:90B; 28–30 min. 10A:90B–80A:20B, flow rate—0.9 mL/min, and detection at 330 nm for all compounds except arctigenin–275 nm. Rutin, myricetin, epicatechin, neochlorogenic, and ferulic acids were determined according to the method previously described by Teneva et al. [68] using a Nexera-i LC-2040C Plus UHPLC system (Shimadzu, Kyoto, Japan), equipped with a UV detector and a binary pump at 280 nm with a sample injection volume of 20 μL. The phenolic compound separation was performed on an Agilent TC-C18 column (5 μm, 4.6 mm × 250 mm) at 25 °C, and the mobile phase included 0.5% acetic acid (A) and 100% acetonitrile (B) at a flow rate of 0.8 mL/min. The compounds of interest in the extracts were identified through their retention times as well as by comparing their absorption spectra with those of standard substances and calculating the correlation coefficients (*r*) between standard spectra and the spectra of the samples based on Equation (1)
(1)$$r=\frac{\sum (a_{i}-\bar{a})(b_{i}-\bar{b})}{\sqrt{\sum (a_{i}-{\left.\bar{a}\right)}^{2}\sum {(b_{i}-\bar{b})}^{2}}}$$
where *a_i_* and *b_i_* are the absorbance values at the *i*th wavelength.

### 4.5. Experimental Animals and Treatment

#### 4.5.1. Experimental Animals

Male Wistar rats with an average weight of 180 to 200 g were used. The animals were kept under standard laboratory conditions—12:12 h dark/light cycle, 45% relative humidity, temperature 26.5 °C ± 1 °C, and free access to food and water. All experimental procedures were conducted in accordance with the Directive 2010/63/EU of the European Parliament and of the Council of 22 September 2010 on the protection of animals used for scientific purposes and Regulation No. 20 of 1 November 2012 on the minimum requirements for the protection and humane treatment of laboratory animals and the requirements for the facilities for their use, breeding, and/or supply issued by the Bulgarian Ministry of Agriculture and Food. The experiments were approved by the Committee on Animal Ethics of the Bulgarian Agency for Food Safety permit № 299 from 15 April 2021 and the decision of the Ethical Committee at MU Plovdiv with protocol № 3 from 20 May 2021.

#### 4.5.2. Treatment

The water-ethanol extracts, obtained according to the description in Section 4.4.1., were filtered and concentrated by evaporation under a vacuum (vacuum evaporator Buchi Labortechnick, Switzerland) until the complete removal of ethanol and the production of aqueous solutions. The amount of concentrated extract that each rat should receive according to its weight was calculated so that it received a dose of 500 mg/kg body weight. To calculate the yield, the concentrated aqueous extract was further dried entirely for one day. The yield obtained was 37.5%.

Forty-eight male Wistar rats were randomly divided into six groups (*n* = 8) as follows: C_0_, C-LPS, EP+LPS–500 mg/kg bw; OA+LPS–500 mg/kg bw, CE1+LPS–500 mg/kg bw, CE2+LPS–500 mg/kg (Table 4). The extract doses were determined based on existing literature data concerning the application of individual extracts in rat models [20,54,69]. Water plant extracts were administered by oral gavage for 14 days before LPS administration. Control groups received saline in a dose of 10 mL/kg bw. Given that *E. purpurea* is a plant with a proven effect on the immune response, it was used in the present study both as a positive control and as an experimental group. A systemic inflammatory response was induced by an intraperitoneal injection of LPS from *E. coli* O55:B5 (250 µg/kg) on day 15 of the experiment. Blood samples for immunological assays were collected 4 h after LPS injection.

#### 4.5.3. Blood Samples

Pyrogen and endotoxin free collecting tubes were used. Blood samples were centrifuged (for 10 min) following clotting. The serum was carefully separated, aliquoted, and frozen at −70 °C.

### 4.6. Immunological Assay

The proinflammatory cytokines TNF-α and INF-γ and the anti-inflammatory cytokine IL-10 in rat serum were tested by commercially available ELISA kits (DiaClone, Besançon, France) in strict compliance with the manufacturer’s guidelines. Principle of the method: for quantitative cytokine testing, rat serums for TNF-α, INF-γ, and IL-10, internal controls, and test standards were dripped on solid phase with monoclonal antibodies against the corresponding cytokine. After incubation and washing, a peroxidase conjugate (second anti-species antibody) is placed to form a cytokine complex. A second wash follows to remove unbound conjugate. When a chromogenic substrate is added to the enzyme, a color reaction occurs, marking the presence of cytokines. Absorption, which is proportional to the cytokine concentration, is measured colorimetrically on a TECAN ELISA reader at 450 nm. The concentration of each cytokine in pg/mL is determined by plotting a standard curve.

### 4.7. Statistical Analysis

Statistics were performed with IBM SPSS Statistics 19.0. All data are expressed as mean ± SEM (standard error of the mean). Data were analyzed by repeated measures of one-way ANOVA, followed by LSD (least significant difference) posthoc test for comparisons between the groups. A value of *p* < 0.05 was considered statistically significant.

## 5. Conclusions

The combinations of extracts result in a richer and more diverse content of bioactive compounds, which interact synergistically, enhancing, complementing, or altering each other’s effects. The results are more pronounced and signify therapeutically advantageous immunomodulatory outcomes compared to the effects of *E. purpurea* and *O. acanthium* extracts alone. The combination of *E. purpurea* and *O. acanthium* in a ratio of 1:1 appears to be more effective in decreasing proinflammatory cytokines. This could be useful in situations where enhancing the immune response against an infectious agent carries the risk of excessive immune system activation, such as cytokine storm. The combination of *E. purpurea* and *O. acanthium* in a ratio of 3:1 is more effective in restoring the balance between TNF-α (Th1 cytokine) and IL-10 (Th2 cytokine). This is associated with a favorable modulation of both inflammation and the immune response while minimizing the risk of immune-mediated adverse reactions. Our findings provide a scientific foundation for further investigation of these unique combined extracts, with the goal of evaluating their potential for clinical application.

## Figures and Tables

**Figure 1 plants-13-03397-f001:**
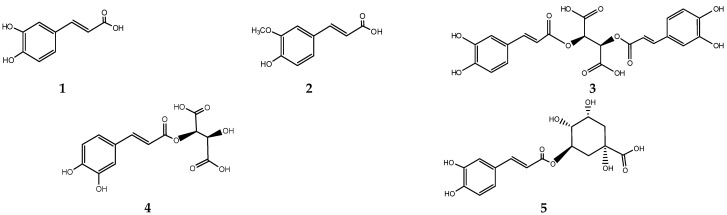
Biologically active substances in the studied extracts—caffeic acid—**1**, caffeic acid derivatives (ferulic acid—**2**, chicoric—**3**, cafratic—**4**, chlorogenic—**5**, neochlorogenic—**6** acids, cynarin—**7**, echinacoside—**8**), flavonoids (apigenin—**9**, quercetin—**10**, myricetin—**11**, epicatechin—**12**, scutellarin—**13**, rutin—**14**), and lignan-arctigenin—**15**.

**Figure 2 plants-13-03397-f002:**
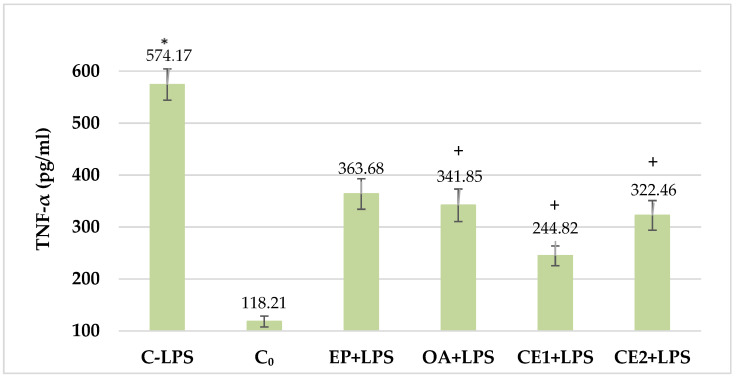
Serum TNF-α in LPS-challenged Wistar rats. The comparison was made using the ANOVA test, followed by LSD posthoc test; * *p* < 0.05 compared to C_0_; + *p* < 0.05 compared to C-LPS.

**Figure 3 plants-13-03397-f003:**
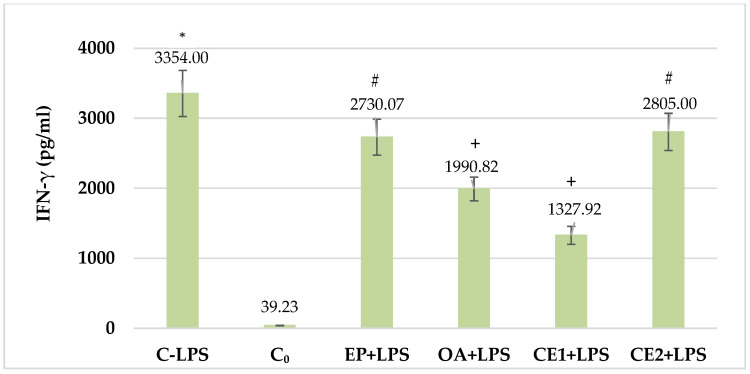
Serum IFN-γ in LPS-challenged Wistar rats. The comparison was made using the ANOVA test, followed by LSD posthoc test; * *p* < 0.05 compared to C_0_; + *p* < 0.05 compared to C-LPS; # *p* < 0.05, compared to CE1+LPS.

**Figure 4 plants-13-03397-f004:**
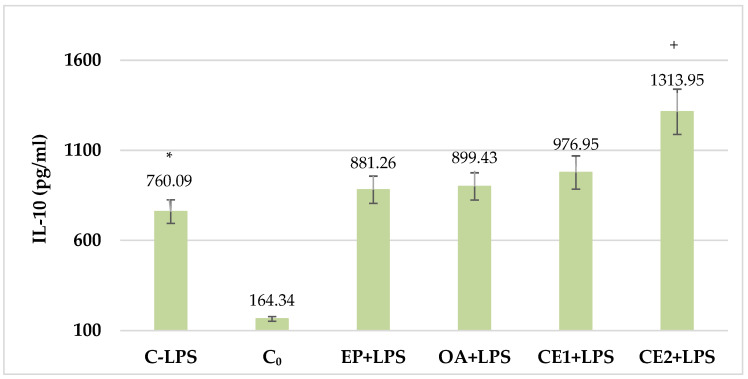
Serum IL-10 in LPS-challenged Wistar rats. The comparison was made using the ANOVA test, followed by LSD posthoc test; * *p* < 0.05 compared to C_0_; + *p* < 0.05 compared to C-LPS.

**Figure 5 plants-13-03397-f005:**
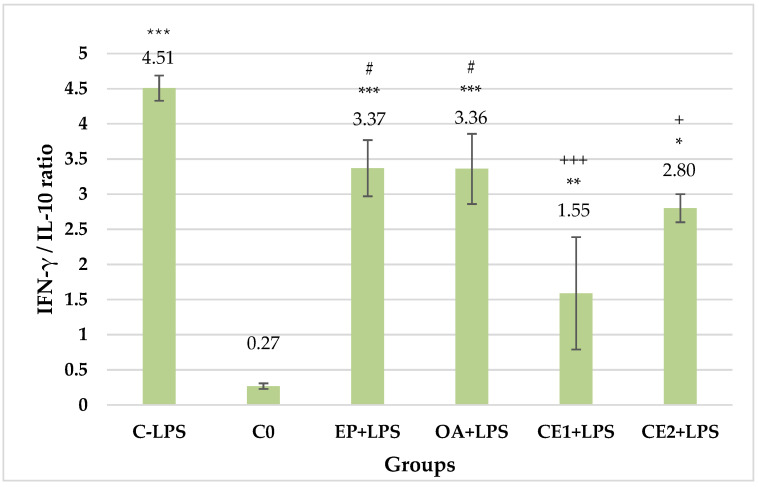
Serum IFN-γ/IL-10 ratio in LPS-challenged Wistar rats. The comparison was made using the ANOVA test, followed by LSD posthoc test; * *p* < 0.05 compared to C_0_; ** *p* < 0.01 compared to C_0_; *** *p* < 0.001 compared to C_0_; + *p* < 0.05 compared to C-LPS; +++ *p* < 0.001 compared to C-LPS; # *p* < 0.05 compared to CE1+LPS.

**Table 1 plants-13-03397-t001:** Content of phenolic and flavonoid compounds in individual extracts of *E. purpurea* and *O. acanthium*, Combination 1, and Combination 2.

Extract	Polyphenols, mg GAE/100 g	Flavonoids, mg QE/100 g
*E. purpurea*	3843.4 ± 34.2	891.0 ± 4.1
*O. acanthium*	1052.2 ± 34.4	186.8 ± 5.0
Combination 1	2833.1 ± 66.4	444.8 ± 4.9
Combination 2	3142.0 ± 18.4	599.1 ± 5.2

Results are presented as mean value ± SD (*n* = 3).

**Table 2 plants-13-03397-t002:** Correlation coefficients (r) of ferulic acid, caffeic acid, caftaric acid, chicoric acid, cynarin, echinacoside, chlorogenic acid, neochlorogenic acid, quercetin, apigenin, rutin, myricetin, epicatechin, scutellarin, and arctigenin.

	Extracts	*E. purpurea*	*O. acanthium*	Combination 1	Combination 2
Substance					
Ferulic acid		0.9987	-	0.9982	0.9984
Caffeic acid		0.9995	0.9998	0.9979	0.9987
Caftaric acid		0.9990	-	0.9986	0.9991
Chicoric acid		0.9985	-	0.9982	0.9994
Cynarin		0.9987	-	0.9994	-
Echinacoside		0.9979	-	-	-
Neochlorogenic acid		0.9994	0.9991	0.9991	0.9989
Chlorogenic acid		0.9981	0.9984	0.9987	0.9992
Quercetin		0.9997	0.9982	0.9986	0.9984
Apigenin		-	0.9981	0.9984	0.9986
Rutin		0.9989	-	0.9993	0.9978
Myricetin		-	0.9979	0.9996	0.9984
Epicatechin		0.9991	0.9983	0.9990	0.9977
Scutellarin		-	0.9977	-	-
Arctigenin		-	0.9994	0.9988	0.9988

**Table 3 plants-13-03397-t003:** Content of biologically active substances in extracts of *E. purpurea*, *O. acanthium*, Combination 1, and Combination 2.

	Extracts	*E. purpurea*	*O. acanthium*	Combination 1	Combination 2
Analyte, μg/g					
Ferulic acid		770.7 ± 44.9	nd	471.9 ± 28.26	726.0 ± 43.56
Caffeic acid		1115.0 ± 55.0	265.0 ± 14.1	696.0 ± 42.9	839.0 ± 24.9
Caftaric acid		3060.0 ± 142.3	nd	1450.0 ± 74.9	2748.0 ± 147.5
Chicoric acid		12,915.7 ± 773.2	nd	6505.0 ± 390.3	8350.0 ± 441.2
Cynarin		39.3 ± 2.1	nd	nd	trace
Echinacoside		55.4 ± 2.3	nd	nd	trace
Neochlorogenic acid		301.0 ± 27.3	596.0 ± 35.3	662.7 ± 49.8	443.8 ± 24.5
Chlorogenic acid		904.7 ± 54.1	661.0 ± 37.3	967.0 ± 55.5	330.6 ± 17.5
Quercetin		270.0 ± 13.3	584.6 ± 33.3	338.0 ± 3.3	98.5 ± 5.1
Apigenin		nd	280.0 ± 3.7	173.2 ± 2.7	57.5 ± 2.9
Rutin		2300.0 ± 132.1	nd	1340.0 ± 76.5	1837.0 ± 115.3
Myricetin		nd	1322.0 ± 66.3	1006.0 ± 65.7	361.0 ± 24.7
Epicatechin		142.3 ± 6.7	139.0 ± 1.2	856.0 ± 51.3	239.0 ± 15.3
Scutellarin		nd	35.0 ± 1.3	trace	nd
Arctigenin		nd	555 ± 32.7	225.2 ± 12.3	108.0 ± 13.2

Results are presented as mean value ± SD (*n* = 3); nd, undetected.

**Table 4 plants-13-03397-t004:** Distribution of experimental animals by groups.

Group	Legend	Description	LPS 250 µg/kg Body Weight
1	C-LPS	Distilled water, 10 mL/kg body weight	Yes
2	C_0_	Distilled water, 10 mL/kg body weight	No
3	EP+LPS	*E. purpurea*, 500 mg/kg body weight	Yes
4	OA+LPS	*O. acanthium*, 500 mg/kg body weight	Yes
5	CE1+LPS	Combination 1, 500 mg/kg body weight	Yes
6	CE2+LPS	Combination 2, 500 mg/kg body weight	Yes

## Data Availability

The data presented in this study are available on request from the corresponding author.

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
