# Peer review of "Echinacea purpurea and Onopordum acanthium Combined Extracts Cause Immunomodulatory Effects in Lipopolysaccharide-Challenged Rats"

_plants, 2024, doi:10.3390/plants13233397_

Round 1
Reviewer 1 Report
Comments and Suggestions for Authors
The observations are described in the attached file.

Author Response
Thank you for giving us the opportunity to submit a revised draft of the manuscript entitled “Echinacea purpurea and Onopordum acanthium combined extracts cause immunomodulatory effect in lipopolysaccharide-challenged rats”.
We are grateful to the reviewers for their comments, recommendations and constructive feedback.
The manuscript draft has been amended as suggested by the reviewers. All the changes in the revised manuscript are highlighted in red.
Reviewer 1
Comment 1: Authors are required to indicate the national or international rule used for the appropriate handling of experimental animals
Response 1: All experimental procedures were conducted in accordance with the Directive 2010/63/EU of the European Parliament and of the Council of 22 September 2010 on the protection of animals used for scientific purposes and Regulation No. 20 of November 1, 2012, on the minimum requirements for the protection and humane treatment of laboratory animals and the requirements for the facilities for their use, breeding, and/or supply issued by Bulgarian Ministry of Agriculture and Food. It is added in the text (lines 423 -429 and 501 – 503).
Comment 2: Several of the chemical structures in Figure 1 are incorrect. For example, both chlorogenic acid and apigenin lack hydroxy groups. In the case of the chemical structures of 1 and 2, either R1 or R2 can be eliminated since H in each case can generate either 1 or 2. Compounds 1 and 2, must be included in the caffeic acid derivatives
Response 2: All structures have been corrected. Corrections are in Figure 1.
Comment 3: In the case of the chemical structures of 1 and 2, either R1 or R2 can be eliminated since H in each case can generate either 1 or 2.
Response 3: Caffeic and ferulic acid formulas have been presented separately.
Comment 4: Compounds 1 and 2, must be included in the caffeic acid derivatives
Response 4: Ferulic acid has been included in the group of caffeic acid derivatives described in the legend of Figure 1.
Comment 5: In pharmacological models, a positive reference is usually used. In this work, an immunomodulatory treatment should have been used to adequately compare the treatments of Echinacea purpurea and Onopordum acanthium.
Response 5: In experimental studies of substances with immunomodulatory effects, there is no established compound that serves as a positive reference. Echinacea purpurea, a plant with well-documented effects on immune function, was utilized in the present study both as positive control and as an experimental treatment. In our study we compared its individual effect on the levels of the studied cytokines with that of combined extracts. This clarification was added to the description of the experimental design.
Comment 6: Although the authors mention that they used commercial samples of the two species, the evidence of their taxonomic identification should be shown.
Response 6: E. purpurea and O. acanthium belong to the Asteraceae family. This information is added on lines 367 and 368. The purchased plant material is accompanied by a quality certificate from MediHerb-83 Ltd., Plovdiv, Bulgaria. This is marked in red (lines 369 – 370).
Comment 7: The authors describe several kinds of extracts generating confusion. In some parts of the text, an aqueous extract is mentioned, in another part an acetone extract is mentioned and the manuscript always refers to an ethanolic extract.
Response 7: The methodology for the determination of total phenols and flavonoids requires extraction with acetone. A 60% ethanol extract was prepared for the chromatographic determination of the individual compounds characteristic of the two plants and for administration to the rats. The same extract was concentrated by evaporating the ethanol under vacuum so that essentially only an aqueous solution was administered to the rats. The amount of concentrated extract that each rat should receive according to its weight was calculated so that it received a dose of 500 mg/kg body weight. To calculate the yield, the concentrated aqueous extract was further dried to dryness for one day. The yield obtained was 37.5%. (Lines 435-438 in the text). A small amount of the extract was used for calculation purposes only, but not for application due to concerns about the stability of the bioactive substances upon heating.
Comment 8: In the extraction process, the quantities of plant material, the volume of solvent, and the way the extract was dried are missing
Response 8: In part 4.4.1. (lines 391-392) it is noted that a 2% solution was prepared. We do not think it necessary to repeat that 2 g were extracted with 100 ml of 60% ethanol.
Comment 9: The yield of each extract is also not mentioned.
Response 9: The yield obtained was 37.5%. Line 439.
Comment 10: The supplementary material containing the chemical identification described in the text is required to complete the review process.
Response 10: The following text is added on lines 412-417, as well as table 2 (in the new version) with the obtained results.
The compounds of interest in the extracts were identified through their retention times as well as by comparing their absorption spectra with those of standard substances and calculating correlation coefficients (r) between standard spectra and the spectra of the samples based on the formula: Lines 416 and 417.
Reviewer 2 Report
Comments and Suggestions for Authors
The present study aimed to combine E. purpurea with O. acanthium, to study the in vivo immunomodulatory effect of two combinations and to compare it with that of the single plants. The experiment showed that combination E. purpurea / O. acanthium in the ratio 1:1 reduced the pro-inflammatory cytokines TNF-α and IFN-γ, and combination E. purpurea / O. acanthium in the ratio 3:1 significantly increased the levels of the anti-inflammatory cytokine IL-10. In conclusion, the obtained results could be used as a basis for future biomedical research about of creating phytopreparations with an immunomodulatory effect. Overall, the topic of this study fully fills in the scope of Plants, but there were several suggestions for further improving the quality of the manuscript.
1. Abstract, several importantly quantified data should be added in this section.
2. Introduction, it is unclear that why the ratios of 1:1 and 3:1 of E. purpurea / O. acanthium were investigated in this study. More background should be added in this section.
3. For the quantification of individual polyphenols, the compounds should be identified by LC-MS.
4. Table 1, the total polyphenols in E. purpurea was 3843.4 ± 34.2, while the total polyphenols in O. acanthium was 1052.2 ± 34.4. However, the content of polyphenols in the combination of E. purpurea / O. acanthium in the ratio 3:1 was 4442.0 ± 18.4, this result is strange, please clarify. In fact, the content of polyphenols in the combination of E. purpurea / O. acanthium with only ratios should be lower than that of the single plant (E. purpurea).
5. Figure 1, please clarify how to identify these compounds in the extract.
6. Table 2 and Table 3 could be combined. Besides, please clarify that why the content of total polyphenols in table 2 and table 3 was extremely lower than that of table 1.
7. Table 4, please clarify that how to select the dosage of extracts.
8. Figures 2-5 could be combined.
9. Too many references were cited in this study.
Author Response
We are grateful to the reviewers for their comments, recommendations and constructive feedback.
The manuscript draft has been corrected as suggested by the reviewers. All the changes in the revised manuscript are highlighted in red.
Comment 1: Abstract, several importantly quantified data should be added in this section.
Response 1: The following text is added in the abstract:
The experiment showed that combination E. purpurea / O. acanthium in the ratio 1:1 re-duced the pro-inflammatory cytokines TNF-α (244.82pg/ml), IFN-γ (1327.92 pg/ml) com-pared to the LPS-control respectively (574.17pg/ml) and (3354.00pg/ml), and combination E. purpurea / O. acanthium in the ratio 3:1 significantly increased the levels of the an-ti-inflammatory cytokine IL-10 (1313.95pg/ml) compared to the LPS-control (760.09pg/ml).
Comment 2: Introduction, it is unclear that why the ratios of 1:1 and 3:1 of E. purpurea / O. acanthium were investigated in this study. More background should be added in this section.
Response 2: The following text is added in the introduction:
We suggest combined extracts of E. purpurea and O. acanthium for the first time. The ratio between the drugs 1:1 was chosen similar to the combination of Echinacea and Hypericum perforatum studied by Bajrai et al /23/, while the ratio 3:1 is entirely our proposal, following the idea of keeping the effect of E. purpurea but eliminating possible side effects.
Comment 3: For the quantification of individual polyphenols, the compounds should be identified by LC-MS.
Response 3: Typically, the identification of compounds can be performed with both MS and DAD detectors. Please see the following link: https://www.chromatographyonline.com/view/peak-purity-liquid-chromatographypart-i-basic-concepts-commercial-software-and-limitations
The following text is added on lines 132-135: The correlation coefficients calculated between the spectra of standards for all these substances and spectra of these compounds in tested samples were between 0.9977 and 0.9998 (Table 2). In this way, the spectral similarity of the peaks of the substances of interest was estimated.
Comment 4: Table 1, the total polyphenols in E. purpurea was 3843.4 ± 34.2, while the total polyphenols in O. acanthium was 1052.2 ± 34.4. However, the content of polyphenols in the combination of E. purpurea / O. acanthium in the ratio 3:1 was 4442.0 ± 18.4, this result is strange, please clarify. In fact, the content of polyphenols in the combination of E. purpurea / O. acanthium with only ratios should be lower than that of the single plant (E. purpurea).
Response 4: Thanks for the remark. A technical error has occurred. The value has been corrected.
Comment 5: Figure 1, please clarify how to identify these compounds in the extract.
Response 5: The following text is added on lines 412-417, as well as Тable 2 (in the new version) with the obtained results. The compounds of interest in the extracts were identified through their retention times as well as by comparing their absorption spectra with those of standard substances and calculating correlation coefficients (r) between standard spectra and the spectra of the samples based on the formula: Lines 416 and 417.
Comment 6: Table 2 and Table 3 could be combined. Besides, please clarify that why the content of total polyphenols in table 2 and table 3 was extremely lower than that of table 1.
Response 6: We have combined Tables 2 and 3 as per your recommendation. The amounts of individual compounds are lower compared to total polyphenols and flavonoids, as we determined only those that are characteristic of both plants. We did not aim to define all possible ones.
Comment 7: Table 4, please clarify that how to select the dosage of extracts.
Response 7: The doses of the extracts are based on available literature data regarding the use of the individual extracts in rat experiments. Reference No. 54 and 20. One more reference, regarding Echinacea purpurea extract dose in immune studies in rats, has been added: Sadigh-Eteghad S, Khayat-Nuri H, Abadi N, Ghavami S, Golabi M, Shanebandi D. Synergetic effects of oral administration of levamisole and Echinacea purpurea on immune response in Wistar rat. Res Vet Sci. 2011;91(1):82-85. It is clarified in section Material and methods 4.5.2. Treatment.
Comment 8: Figures 2-5 could be combined.
Response 8: We prefer that Figures 2-5 remain separate for clarity. Otherwise, the legend will be bulky and complicated.
Comment 9: Too many references were cited in this study.
Response 9: The journal has no limit on the number of references. Furthermore, we believe that a larger number of references also warrants a more in-depth discussion.
Round 2
Reviewer 1 Report
Comments and Suggestions for Authors
Including the supplementary material with chromatograms and UV spectra of the characterized compounds.
Author Response
Thank you for your advice. We have followed your recommendation and added chromatograms ofthe extracts and spectra of the substances of interest as supplementary material.
Reviewer 2 Report
Comments and Suggestions for Authors
The revised manuscript could be accepted.
Author Response
Thank you for your evaluation and recommendations that have contributedto improving the quality of our manuscript.